# Diclofenac Interacts with Photosynthetic Apparatus: Isolated Spinach Chloroplasts and Thylakoids as a Model System

**DOI:** 10.3390/plants13162189

**Published:** 2024-08-08

**Authors:** Monika Majewska, Małgorzata Kapusta, Anna Aksmann

**Affiliations:** 1Department of Plant Experimental Biology and Biotechnology, Faculty of Biology, University of Gdańsk, ul. Wita Stwosza 59, 80-308 Gdańsk, Poland; monika.majewska@ug.edu.pl; 2Bioimaging Laboratory, Faculty of Biology, University of Gdańsk, ul. Wita Stwosza 59, 80-308 Gdańsk, Poland; malgorzata.kapusta@ug.edu.pl

**Keywords:** diclofenac, isolated chloroplasts, photosynthesis, phytotoxicity, *Spinacia oleracea*, thylakoids

## Abstract

Diclofenac, often detected in environmental samples, poses a potential hazard to the aquatic environment. The present study aimed to understand the effect of this drug on photosynthetic apparatus, which is a little-known aspect of its phytotoxicity. Chloroplasts and thylakoids isolated from spinach (*Spinacia oleracea*) were used for this study and treated with various concentrations of diclofenac (from 125 to 4000 μM). The parameters of chlorophyll *a* fluorescence (the OJIP test) as measurements for both the intact chloroplasts and the thylakoid membranes revealed that isolated thylakoids showed greater sensitivity to the drug than chloroplasts. The relatively high concentration of diclofenac that is required to inhibit chloroplast and thylakoid functions suggests a narcotic effect of that drug on photosynthetic membranes, rather than a specific interaction with a particular element of the electron transport chain. Using confocal microscopy, we confirmed the degradation of the chloroplast structure after DCF treatment, which has not been previously reported in the literature. In conclusion, it can be assumed that diclofenac’s action originated from a non-specific interaction with photosynthetic membranes, leading to the disruption in the function of the electron transport chain. This, in turn, decreases the efficiency of photosynthesis, transforming part of the PSII reaction centers into heat sinks and enhancing non-photochemical energy dissipation.

## 1. Introduction

Many biologically active substances are environmental contaminants due to human activity, and one significant class of these contaminants are pharmaceuticals. Non-steroidal anti-inflammatory drugs (NSAIDs) are often used by people to reduce inflammation and to relieve pain, and their amounts in the environment are constantly increasing. One of the most popular NSAIDs, diclofenac (DCF), is detected in freshwater systems over a range from 0 ng/L to 57.16 μg/L [1,2]. DCF is poorly metabolized in the human body, slowly (bio)degraded in the environment [3,4], and, in comparison with other anti-inflammatory drugs (e.g., paracetamol, acetylsalicylic acid, and ibuprofen), it is more toxic to aquatic wildlife [1,5,6]. This is one of the reasons why it has been designated as a major hazardous substance [7].

DCF is used by humans; thus, its mechanism of action in mammalian cells is well understood [8]. However, knowledge regarding the influence of DCF on non-target organisms, especially plants, is limited. Nevertheless, some phytotoxic effects of DCF have been identified. It was shown that DCF disrupts the growth and development of *Zea mays* [9], inhibits the growth of *Lemna minor*, and contributes to a loss of cell membrane integrity in this plant [10,11]. Cell cycle disruption, changes in the pattern of gene transcription, and a reduction in the mitochondrial membrane potential of the DCF-treated unicellular green algae *Chlamydomonas reinhardtii* were also reported [12,13,14]. These kinds of alterations support the suggestion that DCF could cause both cytotoxic and genotoxic effects in plants [15]. However, a significant target of DCF action is believed to be photosynthesis, since the inhibitory effects of many pharmaceuticals on photosynthetic electron transport have been described in the literature. For example, metformin (an anti-diabetic drug) disrupts photosystem II (PSII) functioning and inhibits electron transport beyond PSII, which results in increased non-photochemical energy quenching in *Chlorella vulgaris* [16]. Further, three antibiotics—erythromycin, ciprofloxacin, and sulfamethoxazole—significantly inhibit primary photochemistry and electron transport in *Selenastrum capricornutum* [17]. Some NSAIDs, like ibuprofen, aspirin, and ketoprofen, show the same effect on the alga *Scenedesmus obliquus* [18], with studies by Vannini et al. [6] describing the dose- and time-dependent DCF-induced damage of photosynthetic apparatus in the fern *Azolla filiculoide*. Hájková et al. [10] have demonstrated reduced Rubisco activity in *Lemna minor* under the influence of DCF. DCF was also reported to reduce the content of photosynthetic pigments in cells [6,11,19]. 

Despite the number of studies concerning increases in DCF phytotoxicity, the exact causes of disturbance caused by DCF in photosynthetic apparatus are still unclear. One of these reasons could be that the results obtained for intact plant organisms or algae cells are the sum of all biochemical processes, which makes it difficult to estimate the toxicant’s impacts on individual metabolic pathways. This is especially pronounced in the cases of photosynthesis and respiration because of the similarities between both processes. In both mitochondria and chloroplasts, carbon cycles take place (the carbon-reducing Calvin–Benson–Bassham cycle in chloroplasts and the carbon-oxidizing tricarboxylic acid cycle in the mitochondrion); both organelles contain electron transport chain elements, and both are involved in the production of energy in the form of ATP [20]. This poses the possibility that toxic substances can affect both these organelles, which may cause ambiguous results for cell energetics, carbon flux, and oxygen metabolism, including reactive oxygen species production/scavenging. Moreover, it has been proven that chloroplasts and mitochondria cooperate closely to ensure that photosynthesis is carried out optimally [21,22] and that flexible chloroplast–mitochondrion interactions can overcome major issues in energy metabolism. 

Considering the above information, studies on the effect of DCF on isolated chloroplasts and thylakoids seem to be particularly important, enabling the analysis of this substance’s direct effect on photosynthetic apparatus. Surprisingly, such studies can hardly be found in the literature. Working on isolated *Lemna minor* chloroplasts, Hajkova et al. [10] have shown that DCF can induce the production of reactive oxygen and nitrogen species, leading to the peroxidation of chloroplast lipids and the disruption of photosynthetic processes. However, despite the observed lipid peroxidation, potential changes in the chloroplast structure were not analyzed in this work, and studies on isolated thylakoids conducted were also not conducted, which could shed light on the problem of DCF’s effect on the light-dependent reaction of photosynthesis, which prompted us to undertake the work presented in our paper.

In this study, to separate the influence of DCF on photosynthesis from its influence on other cell organelles, especially mitochondria, we decided to conduct research on isolated chloroplasts and thylakoids from spinach (*Spinacia oleracea*), which is a model organism in photosynthetic research [23,24,25]. Here, chloroplasts and thylakoids were exposed to various concentrations of DCF, and then their activities were assessed using an analysis of chlorophyll *a* fluorescence kinetics (the OJIP test). The OJIP test is one of the most popular techniques for the assessment of pollutant toxicity towards plants, giving precise information regarding the functioning of PSII, which is highly sensitive to environmental stress [26,27]. Further, the imaging of structural changes in DCF-treated chloroplasts was performed using confocal laser scanning microscopy (CLSM), which, to our knowledge, has not been reported in the literature so far. The above-mentioned techniques have allowed us to describe DCF-induced changes in the functioning and structure of photosynthetic apparatus that are potentially responsible for the phytotoxic activity of the drug.

## 2. Materials and Methods

### 2.1. Chemicals and Buffers

Diclofenac sodium salt (DCF, 2-[(2,6-dichlorophenyl)amino]benze-neacetic acid sodium salt; 98% purity) (ABRC, Karlsruhe, Germany) was dissolved in distilled water to obtain a stock solution of a concentration of 10 mg mL^−1^ (31.432 mM). The following buffers were used: Buffer (A): 50 mM HEPES-KOH (pH 7.5) with 0.33 M sucrose, 0.1% BSA, 1 mM MgCl_2_, 1 mM MnCl_2_, and 2 mM EDTA; Buffer (B): 50 mM HEPES-KOH (pH 7.5) with 0.33 M sucrose, 10 mM NaCl, 1 mM MgCl_2_, and 2 mM EDTA; Buffer (C): 50 mM HEPES-KOH (pH 7.5) with 0.33 M sucrose; Buffer (D): 25 mM HEPES buffer (pH, 7.5) with 20 mM NaCl and 5 mM MgCl_2_; Buffer (E): 25 mM HEPES buffer (pH, 7.5), 20 mM NaCl, 5 mM MgCl_2_, and 0.8 M sucrose; Buffer (F): 25 mM HEPES buffer (pH, 7.5), 20 mM NaCl, 5 mM MgCl_2_, and 0.4 M sucrose.

### 2.2. Chloroplast Isolation

Intact chloroplasts were isolated according to Nakano and Asada [28]. Spinach leaves were homogenized through brief blending in Buffer (A). The homogenate was filtered through two layers of etamine and centrifuged (2000× *g*, 4 min, 4 °C). The pellet of crude chloroplasts was resuspended in Buffer (A) and applied to a Percoll gradient prepared in Buffer (B), consisting of four phases (from the top): 10% (*v*/*v*), 3 mL; 40%, 5 mL; 70%, 5 mL; 90%, 5 mL. The samples were then centrifuged at 8000× *g* for 20 min at 4 °C. Intact chloroplasts at the boundaries of layers 70% and 90% were collected, and the chlorophyll concentrations in the layers were measured spectrophotometrically [19]. Further, the chloroplast suspension was diluted with Buffer (C) to obtain a chlorophyll concentration of 50 mg mL^−1^ and then stored at −80 °C. The integrity of isolated chloroplasts was assessed according to the method described by Joly and Carpentier [29]. A chloroplast suspension with an integrity not lower than 80% was used for subsequent experiments. 

### 2.3. Thylakoid Preparation

Thylakoids were prepared based on research by Khorobrykh and Ivanov [30]. The isolated chloroplasts were suspended in Buffer (D) and left for 1 min on ice to induce osmotic shock. Further, an equal volume of Buffer (E) was added to the suspension of broken chloroplasts, and thylakoids were deposited via centrifugation (10 min, 5000× *g*, 4 °C). The pellet was resuspended in Buffer (F) and stored at −80 °C.

### 2.4. Exposure Experiments

The frozen chloroplast suspension was melted on ice in darkness. DCF stock solution was added to the suspension to obtain final concentrations of 125, 250, 500, 1000, 2000, or 4000 µM. As a control, chloroplast suspensions without DCF were prepared. In the first series of the experiments, chloroplast suspensions were incubated with DCF for 5, 10, 15, 30, 45, or 60 min. Based on these results (Appendix A), 15 min incubation time was chosen for the next series of experiments, since extending the incubation time more than 15 min did not increase the effect of DCF on primary chemistry reactions (φP0), which is considered a basic parameter of chlorophyll *a* fluorescence analysis [31,32]. 

The frozen thylakoid suspensions were melted on ice in darkness. DCF stock solution was added to the suspension to obtain final concentrations of 1000, 2000, or 4000 µM and incubated for 15 min. As a control, thylakoid suspensions without DCF were prepared.

### 2.5. Chlorophyll a Fluorescence Measurements (the OJIP Test)

Chlorophyll *a* fluorescence (the OJIP test) was measured using a Handy Pea fluorometer (Hansatech Ltd., Norfolk, UK), as described in [31,32], with modifications. The chloroplast suspension (control or incubated with DCF) was diluted in Buffer (A) to OD_680_ = 0.2, and an aliquot with a volume of 2 mL was used for measurement. The same procedure was applied for a thylakoid suspension diluted with Buffer (F), which was supplemented with 1 mM methyl viologen as an artificial electron acceptor. The following OJIP parameters were analyzed: RC_M_—the fraction of active PSII reaction centers in the sample referring to time, the timepoint at which FM (the maximal fluorescence value under saturating light intensity and a fully reduced QA pool) is reached; the specific energy fluxes per reaction center (RC) for absorption (ABS/RC), trapping (TR_0_/RC), electron transport (ET_0_/RC), and dissipation (DI_0_/RC); the maximum yield of primary photochemistry (φP0); the quantum yield of energy dissipation (φD0); the efficiency of the trapped exciton moving the electron further than QA (ψ0); the maximum yield of electron transport (φE0); and the activity of the water-splitting complex on the donor side of PSII (F_V_/F_0_) [14,19,31]. Details of the protocol applied for the OJIP test and the kinetics of chlorophyll *a* fluorescence transients are outlined in the Appendix A.

### 2.6. CLSM and Image Processing

Specimens were visualized with a Leica STELLARIS 5 WLL (Leica Microsystems, Wetzlar, Germany) confocal microscope with a lightning deconvolution module using laser line 405 nm. The presented images are maximum projections from Z-stacks, which were used to improve the spatial resolution. Four photos were taken from each variant (16 in total), and they are representative of the analyzed samples.

### 2.7. Statistical Analysis

Statistical analysis was performed using MS Excel 2016 (Microsoft Corp., Redmond, WA, USA). All data were expressed as the mean of at least four independent experiments ± SE. To assess the statistical significance of the differences between the values obtained for the control and experimental variants, homogeneity of variances was analyzed, followed by the Student’s *t*-test (for results normal distribution) or Mann–Whitney U-test (if results were not normally distributed), as they are the common tests used in toxicological studies for unpaired two-group comparisons [33]. A value of *p* < 0.05 was considered significant.

## 3. Results and Discussion

Contamination of the environment with non-steroidal anti-inflammatory drugs pushes researchers to look for information regarding their impacts on living organisms. Despite knowledge about one of the most dangerous representatives of this group, DCF, which is mainly focused on mammalian cells [2,34,35], its influences on higher plants and algae are also noticeable [2,36,37]. Since these organisms are the primary producers and they form the basis of trophic chains, changes in their functioning translate into disturbances in the structure of the entire ecosystem [38,39]. The available literature indicates that one of the main causes of DCF phytotoxicity is its adverse effect on photosynthesis, a key process in photoautotrophic cells. DCF’s destructive effect on photosynthesis intensity and photosynthetic pigment content has been described for algae, ferns, lichens, and higher plants [6,11,19]; thus, continuous efforts to elucidate the primary cause of photosynthesis disruption by the drug are being made. The majority of the research in this area refers to the intact plant or algae cell, where strong interdependence between cell compartments and between different biochemical pathways occurs, making an estimation of a toxicant effect on individual physiological processes extremely difficult. Thus, to separate the influence of DCF on the photosynthesis process from its influence on mitochondria and other cell organelles, we decided to conduct research on isolated chloroplasts and thylakoids from spinach. 

Changes in chlorophyll *a* fluorescence parameters in DCF-treated isolated chloroplasts and thylakoids (Figure 1 and Figure 2, Table 1 and Table 2) were convergent with the results obtained for intact higher plants or algae cells, where the intensive non-photochemical dissipation of energy, the diminishment of the quantum efficiency of primary photochemical reactions, and the “silencing” of a fraction of the PSII reaction centers without the inhibition of the remaining active PSII reaction centers were reported [11,14,19]. In the present work, DCF caused a decrease in the quantum efficiency of primary photochemistry (φP0) in isolated spinach chloroplasts, from 5% at 250 µM to 15% at 4000 µM, compared to the control (Table 1, Figure 1). 

Comparably, high DCF concentrations were needed to influence other fluorescence parameters in treated chloroplasts. The maximum electron transport efficiency (φE0) and the quantum efficiency of the use of entrapped energy (ψ0) diminished by 5–20%, and the activity of the water-splitting complex on the donor side of PSII (F_V_/F_0_) was reduced by 30% in chloroplasts treated with DCF in concentrations of 1000, 2000, and 4000 µM (Table 1, Figure 1). On the other hand, the quantum yield of non-photochemical energy dissipation (φD0) increased in the treated chloroplasts by 5% at 250 µM of DCF, up to 25% at 4000 µM of DCF, compared with the control (Table 1, Figure 1). To analyze DCF’s influence on photosynthesis in more detail, the specific energy fluxes through one PSII reaction center were analyzed. It was found that the fraction of active PSII reaction centers (RC_M_) in chloroplasts treated with DCF was reduced by 5% up to 20% with an increasing concentration of the toxicant (Table 1, Figure 1). Nevertheless, these RCs that remained active seemed to function properly, since specific energy absorption, trapping, and electron transport through one active RC did not show significant changes, except for a 10% decrease in TR_0_/RC at concentrations from 1000 µM to 4000 µM, as well as an approximately 30% increase in non-photochemical energy dissipation (DI_0_/RC) (Table 1, Figure 1), which was once again consistent with previous reports in the literature [11,12,14,19].

It must be stressed that the above-mentioned results, analyzed on the background of literature data, indicated the relatively low sensitivity of isolated chloroplasts to DCF, compared to intact algal cells or whole plants. A decrease in PSII activity was noticed in intact *Chlamydomonas reinhardtii* cells exposed to 421.35 µM of DCF [12,14,19] and in *Lemna minor* plants exposed to 0.34 mM of DCF [11]. On the other hand, relatively high concentrations of DCF (1000 mM) were needed to diminish the quantum yield of PSII and enhance non-photochemical energy dissipation in chloroplasts isolated from *Lemna minor* [10]. Such a low sensitivity of isolated chloroplasts compared to intact cells can result from several different reasons. Some authors suggest that in intact cells, it is not only the chloroplast itself but also the chloroplast–mitochondrion crosstalk that is disturbed, leading to cell sensibilization due to the drug’s toxicity. This occurs because efficient chloroplast–mitochondrion cooperation is known to be important for plant stress defense systems [40]. Another concept [10] assumes that the reason for the low sensitivity of isolated chloroplasts can be the relatively short timeframe of DCF-chloroplast incubation to keep the chloroplast highly active. In this case, it is assumed that the amount of DCF crossing the chloroplast envelope is not sufficient to influence the photosynthetic machinery significantly. This concept assumes that the chloroplast envelope is not destroyed by DCF; thus, to verify this assumption, we decided to examine the isolated chloroplasts using confocal microscopy. Our study revealed an unfavorable effect of DCF on the chloroplast membrane after just 15 min of exposure, especially at the highest (4000 µM) drug concentration (Figure 3). Moreover, to allow the DCF molecules to interact directly with the elements of the photosynthetic electron transport chain, experiments were performed with isolated thylakoids. As was expected, the thylakoids were more sensitive than the chloroplasts to DCF. Despite DCF in the concentration of 1000 µM inhibiting PSII functioning in thylakoids by only 5%, an inhibition of 70% was observed after exposing thylakoids to 4000 µM of the drug (Table 2, Figure 2). In the case of the maximum electron transport efficiency (φE0) and the quantum efficiency of the use of entrapped energy (ψ0), DCF-treated thylakoids had these yields reduced by 30% and 80%, respectively (Table 2, Figure 2). DCF-treated thylakoids also showed an increase in the quantum yield of energy dissipation (φD0) from around 140% (2000 µM DCF) to 250% (4000 µM DCF) of the control (Table 2, Figure 2). The fraction of active PSII reaction centers (RC_M_) in isolated thylakoids was reduced by 20% in 1000 µM, by 50% in 2000 µM, and by 90% in 4000 µM of DCF, respectively (Table 2, Figure 2). The activity of the water-splitting complex on the donor side of PSII (F_V_/F_0_) was reduced to the same level by 15, 40, and 90% in the thylakoids treated with DCF at concentrations of 1000, 2000, and 4000 µM, respectively (Table 2, Figure 2). This strongly supports the suggestion that heat sink-PSII reaction centers are created because such PSIIs are characteristic of the reaction centers in which both the donor and acceptor sides are inhibited [25]. Due to the diminishment of the active PSII RC’s fraction, the calculated values of parameters for the specific energy fluxes through one RC significantly increased (Table 2, Figure 2). 

The relatively high DCF concentration required for the functional inhibition of isolated chloroplasts and thylakoids suggests a narcotic action of the drug on the photosynthetic membranes rather than a specific interaction with a particular electron transport chain element. Indeed, the data within the literature indicate that the phytotoxic effect of pharmaceuticals is mainly manifested via narcosis [41,42] and that membrane degradation by various drugs is often reported in the literature [43,44,45]. Furthermore, there is consensual evidence that the lipophilicity of NSAIDs is one of the most important factors behind their toxic action [43,44,45]. For example, Ferreira et al. [43] have proven that DCF can be incorporated into the phospholipid layers of phosphatidylcholine (EPC) liposomes, which are used as cell membrane models, with the drug preferentially localizing at the surface of the membrane close to the phospholipid headgroups. Further, membrane breakdown by DCF has been demonstrated in *Lemna minor* using the quantification of lipid peroxidation end-products and the spectrophotometric Evans blue uptake assay as indirect determinants of the degree of membrane destruction [11]. Because the direct visualization of DCF’s interactions with chloroplasts has not been reported in the literature so far, CLSM was used in the present work to demonstrate the degradation of chloroplast structures caused by DCF at various concentrations. The lowest DCF concentration used in experiments (125 µM) did not show significant changes in chloroplast structure (Figure 3B): they still possessed a regular shape and exhibited strong thylakoid grana fluorescence, comparable to the control chloroplasts (Figure 3A). This observation is consistent with the OJIP test parameters, indicating that DCF-treated chloroplasts were not significantly different from the control (Table 1, Figure 1). However, as the DCF concentration increased (above 1000 µM), the chloroplasts became irregular, and the grana seemed to become disorganized and showed much weaker levels of fluorescence (Figure 3C,D). At the highest (4000 µM) drug concentration, the chloroplast envelopes were clearly damaged, releasing the stroma outside and opening the possibility of direct DCF–thylakoid interactions (Figure 3D). Once again, these results were consistent with chlorophyll *a* fluorescence in vivo parameters, indicating increases in photosynthetic disturbance as the DCF concentration increased (Table 1, Figure 1).

Biomembranes serve as a diffusion barrier that provides cellular compartmentation and enables proper functioning of all organelles. Thus, the integrity and proper fluidity of chloroplast envelopes and thylakoid membranes are crucial for photosynthesis to be efficient. It was shown that for photosynthesis to be efficient, membrane fluidity must remain within relatively narrow limits. The importance of maintaining this fluidity for the plant is demonstrated by the adaptive processes responding to various abiotic stresses, which involve changes in the lipid composition to maintain constant membrane fluidity [46]. It was shown that *Chlamydomonas reinhardtii* mutants lacking sulfo-quinovosyl diacylglycerol, one of the main components of thylakoid membranes, showed a 40% decrease in PSII activity [47]. Further, the reduced photosynthetic efficiency caused by the collapse of thylakoid membranes after exposure of *Prorocentrum minimum* to the herbicide metazachlor was demonstrated by Kim et al. [48]. When the integrity of the thylakoid membrane is invaded, the photosynthetic protein complexes are unstable and photosynthetic activity is limited. Thus, it can be assumed that DCF’s action originated from a non-specific interaction with photosynthetic membranes, leading to the disruption of electron transport chain functioning. This, in turn, decreased the photosynthesis efficiency, transforming part of the PSII RCs into heat sinks and enhancing non-photochemical energy dissipation. It should be noted that disturbances in light-dependent photosynthetic reactions translate into a reduced efficiency of inorganic carbon assimilation, which leads to the inhibition of the synthesis of sugars and other organic compounds in the long term, ultimately reducing plant productivity [49].

The above considerations strongly indicate that the main cause of DCF phytotoxicity reported for plant organisms [9,10,11] is its destructive effect on photosynthetic apparatus. Taking into account that not only photosynthesis itself but also chloroplast and mitochondria cooperation are inevitable for the proper growth and development of individual plant organisms, their populations, and entire ecosystems [21,22,49], contamination of the environment with DCF poses a serious threat to maintaining an environmental balance and the proper structure of food chains. 

## 4. Conclusions

The most important conclusion from our research is that DCF affects photosynthetic apparatus non-specifically through the destruction of chloroplast membranes, consequently leading to the inhibition of photosynthesis. The destruction of thylakoid membranes by DCF disturbed the flow of electrons in the photosynthetic chain, which resulted in the transformation of some reaction centers into heat radiators to protect photosystems from damage. To reduce the photosynthetic activity in isolated chloroplasts, higher concentrations of DCF were needed as compared to thylakoids, which could have resulted from the protective outer chloroplast membrane preventing DCF from reaching the photosynthetically active thylakoids. 

Since chloroplast membranes are similar in structure to the membranes surrounding other cell organelles and the entire protoplast, the demonstration of a non-specific, destructive effect of DCF on photosynthetic apparatus allows us to assume that similar changes also apply to other metabolic processes. A proposed direction for further research is the analysis of the interaction of DCF with plant mitochondria, whose functioning is as important for the functioning of the plant cell as the functioning of the chloroplast. Such analyses should include studies using an isolated mitochondrial fraction and in vivo studies, including the mitochondria–chloroplast crosstalk, to obtain a broad picture of the effect of DCF on plant cell bioenergetics. In the future, extending this type of analysis to other NSAIDs will contribute to a comprehensive description of the threats resulting from environmental pollution with medicinal substances and enable researchers to propose ways to reduce this threat.

## Figures and Tables

**Figure 1 plants-13-02189-f001:**
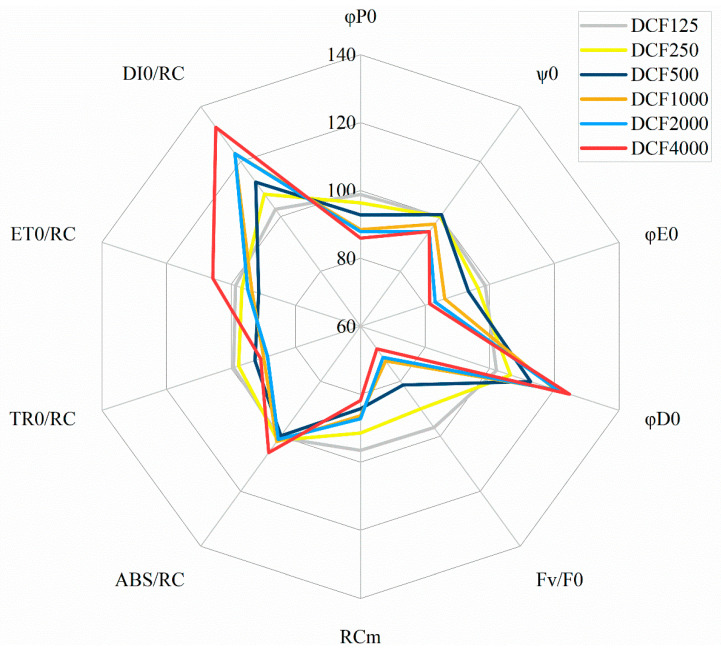
The effect of diclofenac (DCF) at concentrations of 125, 250, 500, 1000, 2000, or 4000 µM on selected OJIP parameters in *Spinacia oleracea* chloroplasts. Compared with the control, DCF diminished the maximum electron transport efficiency (φE0), the quantum efficiency of the use of entrapped energy (ψ0), the activity of the water-splitting complex on the donor side of PSII (F_V_/F_0_), the fraction of active PSII reaction centers (RC_M_), and specific energy trapping (TR_0_/RC) but enhanced the quantum yield of non-photochemical energy dissipation (φD0) and specific energy flux for non-photochemical energy dissipation (DI_0_/RC). Specific energy absorption (ABS/RC), trapping (TR_0_/RC), and electron transport (ET_0_/RC) remained at the level of the control. All values are means (*n* = 4) and are presented as a percentage of the control.

**Figure 2 plants-13-02189-f002:**
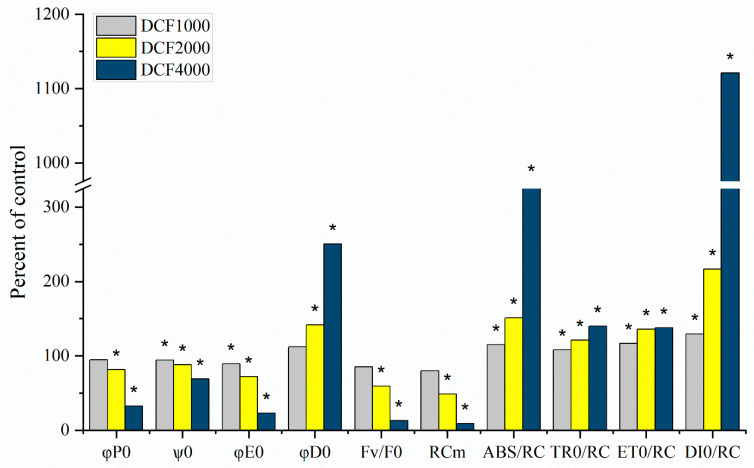
The effect of diclofenac (DCF) at concentrations of 1000, 2000, or 4000 µM on selected OJIP parameters in *Spinacia oleracea* thylakoids. DCF diminished the maximum yield of primary photochemistry (φP0), the maximum electron transport efficiency (φE0), the quantum efficiency of the use of entrapped energy (ψ0), the fraction of active PSII reaction centers (RC_M_), and the activity of the water-splitting complex on the donor side of PSII (F_V_/F_0_), while the quantum yield of energy dissipation (φD0) and specific energy fluxes through one RC (absorption, ABS/RC; trapping, TR_0_/RC; electron transport, ET_0_/RC; and dissipation, DI_0_/RC) significantly increased. Values are means (*n* = 4) and are presented as a percentage of the control. Asterisks indicate the significance of differences (*t*-test, *p* < 0.05) between the control (C) and treated thylakoids.

**Figure 3 plants-13-02189-f003:**
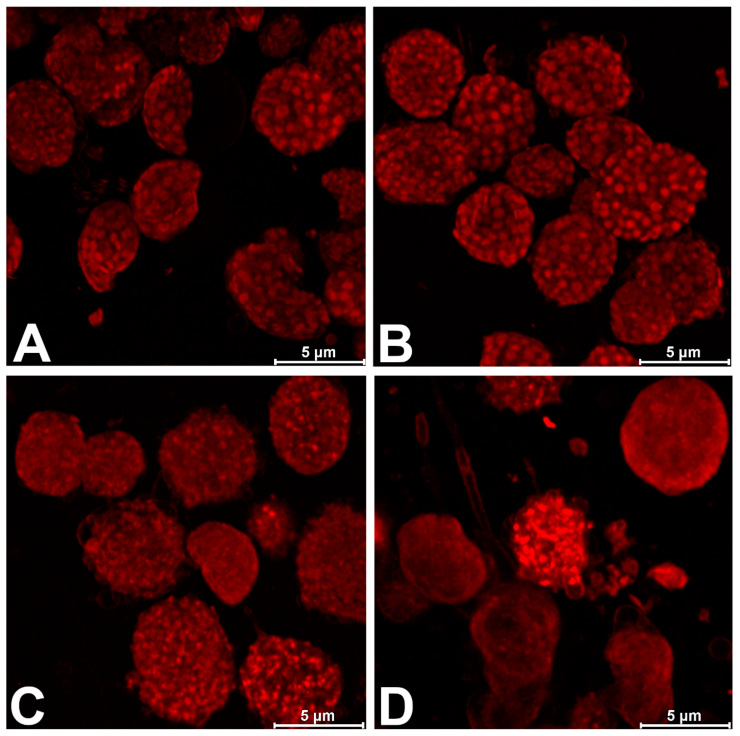
*Spinacia oleracea* chloroplasts imaged using confocal laser scanning microscopy (CLSM) after 15 min of exposure to diclofenac (DCF). Under control conditions (**A**), chloroplasts had a clearly defined, regular grana with autofluorescence typical of these structures. DCF at concentration of 125 µM (**B**) did not change chloroplast structure and enhanced grana autofluorescence. DCF at concentration of 1000 µM (**C**) caused chloroplast irregularity, grana disorganization, and autofluorescence decrease. At the highest (4000 µM) DCF concentration (**D**), the chloroplast envelopes were clearly damaged, the stroma were released outside, grana were almost indistinguishable, and autofluorescence was weak. The presented images are representative of the analyzed samples (*n* = 4).

**Table 1 plants-13-02189-t001:** The effect of diclofenac (DCF), when applied at concentrations of 125, 250, 500, 1000, 2000, or 4000 µM, on the chlorophyll *a* fluorescence parameters of *Spinacia oleracea* chloroplasts. Values are means (*n* = 4) ± SE. Asterisks indicate the significance of differences (*t*-test, *p* < 0.05) between the control (C) and treated chloroplasts. RC_M_—the fraction of active PSII reaction centers; the specific energy fluxes per reaction center for absorption (ABS/RC), trapping (TR_0_/RC), electron transport (ET_0_/RC), and dissipation (DI_0_/RC); φP0—the maximum yield of primary photochemistry; φD0—the quantum yield of energy dissipation; ψ0—the efficiency of the trapped exciton moving the electron further than QA; φE0—the maximum yield of electron transport; F_V_/F_0_—the activity of the water-splitting complex on the donor side of PSII.

	φP0	ѱ0	φE0	φD0	ABS/RC	TR_0_/RC	ET_0_/RC	DI_0_/RC	F_V_/F_0_	RC_M_
C	0.64 ± 0.006	0.71 ± 0.004	0.45 ± 0.004	0.36 ± 0.006	2.39 ± 0.032	1.52 ± 0.021	0.60 ± 0.012	0.87 ± 0.019	1.91 ± 0.047	637 ± 15.21
DCF125	0.63 ± 0.005	0.71 ± 0.003	0.44 ± 0.004	0.37 ± 0.005	2.40 ± 0.013	1.51 ± 0.007	0.59 ± 0.008	0.89 ± 0.017	1.85 ± 0.044	615 ± 13.08
DCF250	0.61 * ± 0.008	0.71 ± 0.004	0.43 ± 0.007	0.39 * ± 0.008	2.43 ± 0.027	1.49 ± 0.012	0.58 ± 0.010	0.94 ± 0.029	1.72 * ± 0.060	582 * ± 17.62
DCF500	0.59 * ± 0.008	0.71 ± 0.004	0.42 * ± 0.006	0.41 * ± 0.008	2.39 ± 0.020	1.41 * ± 0.018	0.55 * ± 0.012	0.98 * ± 0.023	1.55 * ± 0.051	537 * ± 16.08
DCF1000	0.56 * ± 0.008	0.69 ± 0.008	0.39 * ± 0.009	0.44 * ± 0.008	2.44 ± 0.045	1.37 * ± 0.015	0.56 ± 0.014	1.07 * ± 0.038	1.38 * ± 0.044	550 * ± 11.62
DCF2000	0.56 * ± 0.006	0.67 * ± 0.014	0.37 * ± 0.012	0.44 * ± 0.006	2.42 ± 0.039	1.35 * ± 0.012	0.57 ± 0.014	1.07 * ± 0.031	1.36 * ± 0.033	555 * ± 9018
DCF4000	0.55 * ± 0.009	0.67 * ± 0.013	0.37 * ± 0.012	0.45 * ± 0.009	2.53 * ± 0.044	1.38 * ± 0.011	0.64 ± 0.012	1.15 * ± 0.041	1.30 * ± 0.046	521 * ± 7.88

**Table 2 plants-13-02189-t002:** The effect of diclofenac (DCF), when applied at concentrations of 1000, 2000, or 4000 µM, on the chlorophyll *a* fluorescence parameters in *Spinacia oleracea* thylakoids. Values are means (*n* = 4). Asterisks indicate the significance of differences (*t*-test, *p* < 0.05) between the control (C) and treated thylakoids. RC_M_—the fraction of active PSII reaction centers; the specific energy fluxes per reaction center for absorption (ABS/RC), trapping (TR_0_/RC), electron transport (ET_0_/RC), and dissipation (DI_0_/RC); φP0—the maximum yield of primary photochemistry; φD0—the quantum yield of energy dissipation; ψ0—the efficiency of the trapped exciton moving the electron further than QA; φE0—the maximum yield of electron transport; F_V_/F_0_—the activity of the water-splitting complex on the donor side of PSII.

	φP0	ѱ0	φE0	φD0	ABS/RC	TR_0_/RC	ET_0_/RC	DI_0_/RC	F_V_/F_0_	RC_M_
C	0.69 ± 0.011	0.59 ± 0.007	0.41 ± 0.009	0.31 ± 0.011	2.05 ± 0.050	1.41 ± 0.015	0.58 ± 0.015	0.64 ± 0.038	2.53 ± 0.139	914 ± 52.18
DCF1000	0.65 ± 0.016	0.56 * ± 0.009	0.36 * ± 0.012	0.35 ± 0.016	2.36 * ± 0.094	1.53 * ± 0.024	0.68 * ± 0.024	0.83 * ± 0.071	2.16 ± 0.157	729 ± 63.68
DCF2000	0.56 * ± 0.023	0.52 * ± 0.011	0.29 * ± 0.018	0.44 * ± 0.023	3.10 * ± 0.173	1.72 * ± 0.025	0.79 * ± 0.020	1.38 * ± 0.150	1.50 * ± 0.138	448 * ± 47.46
DCF4000	0.23 * ± 0.020	0.41 * ± 0.020	0.09 * ± 0.012	0.77 * ± 0.020	9.13 * ± 0.761	1.98 * ± 0.041	0.80 * ± 0.043	7.15 * ± 0.772	0.33 * ± 0.036	85 * ± 9.065

## Data Availability

The original contributions presented in this study are included in this article/the Appendix A. The dataset is also published in the open repository Mendeley Data (doi: 10.17632/5yt8gvwhjr.1).

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
