# Peer review of "Diclofenac Interacts with Photosynthetic Apparatus: Isolated Spinach Chloroplasts and Thylakoids as a Model System"

_plants, 2024, doi:10.3390/plants13162189_

Round 1

Reviewer 1 Report

Comments and Suggestions for Authors

The authors investigated the phytotoxicity of DCF, a typical non-steroidal anti-inflammatory drug found in the environment. The authors used biochemically prepared intact chloroplast as a material to evaluate the effect of DCF on chloroplast. Previous and present studies pointed out that photosynthesis should be damaged by adding DCF, and the authors investigated the effect intensively based on chlorophyll fluorescence parameters. The authors described the results precisely, and the subject is essential for stressed science. However, the following concerns will improve the quality of the study.

The question is whether the DCF effect on photosynthesis is reversible or irreversible.

In addition, the data lack the time course of the DCF treatment.

The authors did not provide chlorophyll fluorescence kinetic data (when they obtained OJIP parameters), which is essential for interpreting the data and for an intensive discussion in the future.

Author Response

For the response to the Reviewer comments, please see the attachment.

Reviewer 2 Report

Comments and Suggestions for Authors

The manuscript investigates the effects of diclofenac (DCF), a common environmental drug, on the photosynthetic apparatus using chloroplasts and thylakoids isolated from spinach (Spinacia oleracea) as a model system. Different concentrations of DCF were tested and the study assessed the effects on chlorophyll a fluorescence parameters using the OJIP assay and structural changes using confocal microscopy.

The study thus addresses a relatively unexplored area of diclofenac phytotoxicity, in particular its effects on the photosynthetic apparatus.

The use of isolated chloroplasts and thylakoids allows a more focused examination of the effects of DCF on photosynthesis, eliminating interference from other cellular processes.

The manuscript includes a detailed analysis of chlorophyll a fluorescence kinetics and structural changes by confocal microscopy.

The manuscript is very clearly written and well edited. The introduction provides a good background, but should explicitly state the knowledge gap that this study addresses.

The statistical methods used are mentioned, but the details are insufficient. For example, the exact tests performed, the rationale for their selection and any adjustments for multiple comparisons to strengthen the credibility of the results should be included.

The results are well presented with appropriate use of figures and tables. However, some figures could benefit from more descriptive captions to make them self-explanatory. In particular, in Figure 2, the authors should better specify the changes observed in chloroplast structures after diclofenac treatment. In Figure 4, the changes in chlorophyll a fluorescence parameters (OJIP test) between treated and untreated samples should be explained.

The discussion could be more focused on potential implications for plant health and ecosystem dynamics.

The conclusion is concise, but could reiterate the main findings and their importance more strongly. The authors should also include a brief mention of possible future research directions, which would be helpful.

Overall, the manuscript presents relevant and well structured research on the phytotoxic effects of diclofenac on the photosynthetic apparatus. However, improvements in methodological clarity, discussion of results, and linkage to practical applications would significantly increase the quality and impact of the work.

Author Response

(The authors gave the same response as above.)

Round 2

Reviewer 1 Report

Comments and Suggestions for Authors

The authors responded to the issue of irreversible damage to photosynthesis function by DCF. While the authors presented the time course in the revised manuscript in response to the comment, the provided dataset makes it difficult to read the trend against DCF exposure time. Some graphical presentations such as scatter plots or some corresponding could be improved to interpret the trend for DCF exposure time.

Author Response

Comment 1. The authors responded to the issue of irreversible damage to photosynthesis function by DCF. 

Response 1. Thank you ver much for your comment. We hope that you will find our explanation as acceptable.

Comment 2. While the authors presented the time course in the revised manuscript in response to the comment, the provided dataset makes it difficult to read the trend against DCF exposure time. Some graphical presentations such as scatter plots or some corresponding could be improved to interpret the trend for DCF exposure time.

Response 2: Thank you for drawing out attention to this problem. The manuscript has been supplemented with additional figure (Figure S3 in Supplementary Materials), as recommended. The appropriate reference to the Figure S3 has been added in the main text.